# Inhibiting ALK2/ALK3 Signaling to Differentiate and Chemo-Sensitize Medulloblastoma

**DOI:** 10.3390/cancers14092095

**Published:** 2022-04-22

**Authors:** Doria Filipponi, Marina Pagnuzzi-Boncompagni, Gilles Pagès

**Affiliations:** 1Biomedical Department, Centre Scientifique de Monaco, 98000 Monaco, Monaco; dfilipponi@centrescientifique.mc (D.F.); mpagnuzzi@centrescientifique.mc (M.P.-B.); 2LIA ROPSE, Laboratoire International Associé Université Côte d’Azur—Centre Scientifique de Monaco, 06100 Nice, France; 3University Nice Cote d’Azur, Institute for Research on Cancer and Aging of Nice (IRCAN), CNRS UMR 7284, INSERM U1081, Centre Antoine Lacassagne, 06189 Nice, France

**Keywords:** cancer stem cell, differentiation therapy, BMP4 signaling pathway

## Abstract

**Simple Summary:**

Many cancers re-emerge after treatment, despite the sensitivity of the bulk of tumor cells to treatments. This observation has led to the ‘cancer stem cell’ (CSCs) hypothesis, stating that a subpopulation of cancer cells survive therapy and lead to tumor relapse. However, the lack of universal markers to target CSCs is the main constraint to fully eradicate the CSC pool. Differentiation therapy (DT) might in principle suppress tumorigenesis through conversion of undifferentiated cancer cells of high malignancy into differentiated cells of low tumorigenic potential. Here, we provide evidence that CSCs of medulloblastoma can be forced to resume their differentiation potential by inhibiting the BMP4–ALK2/3 axis, providing a new entry point for medulloblastoma treatment.

**Abstract:**

Background: Medulloblastoma (MB) is a malignant pediatric brain tumor, and it represents the leading cause of death related to cancer in childhood. New perspectives for therapeutic development have emerged with the identification of cancer stem cells (CSCs) displaying tumor initiating capability and chemoresistance. However, the mechanisms responsible for CSCs maintenance are poorly understood. The lack of a universal marker signature represents the main constraints to identify and isolate CSCs within the tumor. Methods: To identify signaling pathways promoting CSC maintenance in MB, we combined tumorsphere assays with targeted neurogenesis PCR pathway arrays. Results: We showed a consistent induction of signaling pathways regulating pluripotency of CSCs in all the screened MB cells. BMP4 signaling was consistently enriched in all tumorsphere(s) independently of their specific stem-cell marker profile. The octamer-binding transcription factor 4 (OCT4), an important regulator of embryonic pluripotency, enhanced CSC maintenance in MBs by inducing the BMP4 signaling pathway. Consistently, inhibition of BMP4 signaling with LDN-193189 reduced stem-cell traits and promoted cell differentiation. Conclusions: Our work suggests that interfering with the BMP4 signaling pathway impaired the maintenance of the CSC pool by promoting cell differentiation. Hence, differentiation therapy might represent an innovative therapeutic to improve the current standard of care in MB patients.

## 1. Introduction

Medulloblastoma (MB) is one the most aggressive tumors of the central nervous system, accounting for approximately 25% of all pediatric brain tumors [1,2]. Single-cell transcriptomic analyses have shown a huge biological and clinical heterogeneity including four major molecular groups: WNT, SHH, group 3, and group 4 [3,4,5]. Among these groups, approximately 50% patients with group 3 MB (characterized by c-MYC upregulation or amplification) are metastatic at the time of diagnosis [6,7] and, therefore, have the worst outcome. The currently available therapy for MB consists of a maximal safe resection, chemotherapy, and radiotherapy (for children ≥3 years old). The overall 5 year survival is approximately 70% in children with primary disease. Relapse occurs in around 30% of children and is fatal in 100% of cases. Only a small subset of tumor cells can initiate tumor and drive relapse. These rare stem cells, called cancer stem cells (CSCs), possess tumorigenic capability with a high proliferation rate, self-renewal, differentiation, and metastatic potential [8,9]. To date, CSCs with high oncogenic potential have been identified and isolated in several types of human cancers, including the brain, [8,10,11]. Current radio- and chemotherapies efficiently kill the bulk of cancer cells but spare a relevant fraction of CSCs. Due to their capability to initiate and support tumorigenesis, CSCs can recapitulate a new heterogeneous tumor and give rise to relapse [12,13]. Several transcription factors involved in embryonic stem-cell maintenance such as the octamer-binding transcription factor 4 (OCT4, also known as OCT3 or OCT3/4), along with NANOG, c-MYC, KFL4, and SOX2, have been identified as critical regulators of gene expression necessary to maintain pluripotency and self-renewal in both human and mouse somatic cells [14,15,16]. In addition, transient ectopic expression of such transcription factors is sufficient to reprogram somatic cells back to pluripotency [17]. Among these factors, OCT4 has emerged as the master transcription factor governing self-renewal and pluripotency in stem cells [18]. OCT4 is also highly expressed in CSCs [19,20,21]. High OCT4 levels drive tumorigenicity and metastatic potential of MB cells [22,23,24]. Consistently, patients with tumors expressing high levels of OCT4 displayed shorter overall survival [25]. In the context of patient risk stratification, it was sufficient to shift patients from the low- to the high-risk groups [25]. Subsequently, high expression of OCT4 correlates with chemoresistance in several types of cancers [26,27]. In addition, several signaling pathways regulating survival, proliferation, self-renewal, and differentiation properties of normal stem cells are important regulators of CSCs [28]. However, the mechanistic link among OCT4 expression, induction of signaling promoting self-renewal, and drug resistance has not yet been delineated in MB. As a such, studies on the downstream activation of OCT4 in conjunction with drug resistance could provide new insights into signaling pathways that support CSCs. In this study, we aimed to identify neurogenesis-related signaling pathways that support CSCs maintenance in MB and promote resistance to standard-of-care therapies. These findings highlight the involvement of BMP4 signaling pathway in a mechanism supporting stemness providing new therapeutic entry points to target CSCs pool in MB patients. 

## 2. Materials and Methods

### 2.1. Cell Lines and Cell Culture

The human MB cell lines (DAOY, HD-MB03, D458) were purchased from American Type Culture Collection (ATCC). DAOY and D458 cells were maintained in DMEM (Gibco, Life Technologies Corporation, Loughborough, UK) supplemented with 7.5% fetal bovine serum (FBS, SIGMA, Burlington, MA, USA). HD-MB03 cells were maintained with RPMI 7.5% fetal bovine serum (FBS, SIGMA, Burlington, MA, USA). Cells were monitored routinely, and the absence of mycoplasma was verified monthly using the PlasmoTest kit (Invivogen, San Diego, CA, USA). 

### 2.2. Sphere-Forming Assays

MB stem-cell cultures were maintained in DMEM/F/12 medium (GIBCO, Life Technologies Corporation, Loughborough, UK) supplemented with B27 and N2 supplement (Thermo Fisher SCIENTIFIC, Waltham, MA, USA), 0.6% Pen–Strep, 2 ng/mL heparin, 20 ng/mL of human recombinant epidermal growth factor (EGF) and basic fibroblast growth factor (bFGF) (EGF from Sigma and bFGF from STEMCELL TECHNOLOGIES, Vancouver, BC, Canada). After 3–5 days of culture in hydrophobic dish at 37 °C with 5% CO2, primary spheres were obtained. MB stem-like cells were dissociated using acutase (Gibco, Life Technologies Corporation, Loughborough, UK) and seeded in a six- or 96-well plate at the density of 60,000 or 3000 cells/mL respectively, depending on experiment. The MB dissociated stem-like cells were then exposed to BMP4 (Sigma-Aldrich, St. Louis, MO, USA) 50 ng/mL or LDN-193189 (STEMCELL Technologies, Vancouver, BC, Canada) 30 nM. We observed the spheres after 72 h. Spheres >75 μm in diameter were counted after 6 days by light microscopy to evaluate the efficacy of their formation. Each experiment was repeated at least three times with three independent wells using ultralow-attachment plates (Corning Inc.; Corning, NY, USA) or the agar-based nonadherent three-dimensional method [29]. The trypan blue assay was used to evaluate cell viability. 

### 2.3. Transient and Lentivirus Transfections

For transient overexpression, cultured MB cells were seeded at 30–40% confluency. Then, 24 h later, cells were transfected with vectors pLenti-CMV-V5-Luc (Addgene plasmid 21474, Watertown, MA, USA) and pSin-EF2-Oct4-Pur (Addgene plasmid 16579) using PEI transfection method. Next, 48 h post transfection, cells were harvested and processed for RNA, proteins, immunofluorescence, FACS analysis, and sphere-forming assay. For viral production, pLenti-CMV-V5-Luc (Addgene plasmid 21474) or pSin-EF2-Oct4-Pur were co-transfected with lentivirus packaging vector psPAX2 (Addgene plasmid 12260) or pMD2.G (Addgene plasmid 12259) into HEK293T cells supplemented with 2 μg/mL polybrene (Sigma-Aldrich) through PEI transfection. Supernatant containing viral particles was recovered after 48 h from HEK293T transfected cells filtered by a 0.45 μm pore membrane and used to infect MB cells (1:10 *v*:*v*) at 37 °C overnight. Medium was then replaced with fresh medium after 48 h, and cells were harvested and processed for RNA, protein, immunofluorescence, FACS analysis, and sphere-forming assay.

### 2.4. Immunoblot

Cells were lysed with 1× Laemmli buffer (6 mM Tris-Cl pH 6.8, 2% SDS, 10% glycerol, 5% β-mercaptoethanol. Protein amounts were determined by the Pierce TM BCA Protein Assay Kit (Thermo Fisher). Then, 50 μg of protein was used for SDS-PAGE. The proteins were transferred onto PVDF membranes in Tris–glycine buffer (25 mM Tris, 192 mM glycine, pH8.3) + 10% ethanol (*v*:*v*). Membranes were blocked with 5% milk at room temperature for 1 h, and then immunoblotted overnight diluted in 3% milk at 4 °C. Membranes were washed three times with PBS–Tween 0.1% and incubated with HRP-conjugated secondary antibodies at room temperature for 1 h. After three final washes in PBS–Tween 0.1%, the Advansta Western Bright Quantum HRP substrate was used as a detection reagent.

### 2.5. Flow Cytometry Analyzes

Cells were dissociated with acutase and washed once with PBS. Cell surface levels were determined with antihuman antibodies CD133-FITC, (Miltenyi Biotec, Solothurn, Switzerland) diluted 1:300 in FACS buffer for 30 min in ice. All samples were analyzed on an FACS Melody (BD Biosciences, Allschwil, Switzerland) using the FlowJo software.

### 2.6. MTT Assays

The details of this assay were described previously [30]. Briefly, 25,000–30,000 cells were seeded in 96-well micrometer plates in relative medium, DMEM 7% FBS for DAOY and D458 and RPMI 7% FBS for HD-MB03. The cells were incubated overnight for attachment. Drug concentrations in serial dilution from 0 to 200 μM were added in triplicate and incubate for 48 h in 5% CO2 at 37 °C. Cells were incubated with MTT solution (5 mg/mL) for 4 h and subsequently treated with DMSO. Absorbance was measured at 570 nm using a 96-well microplate reader (MULTISKAN FC-Thermo Fischer SCIENTIFIC). The percentage cytotoxicity was calculated using untreated cells to correct for background absorbance [30].

### 2.7. qRT-PCR

First, 500 ng of total RNA was reverse-transcribed using the Maxima cDNA synthesis kit with dsDNA (Thermo Fischer) and treated with DNase at 37 °C for 5 min according to manufacturer instruction (Thermo Fisher). Quantitative real-time PCR reactions were then performed using Syber Premix Ex Taq (Ti Rnase H Plus) ROX Plus (Ozyme, Saint-Cyr-l’École FRANCE) in a MicroAmp Optical 96-well plates using StepOne-Plus System (Applied Biosystems, Zug, Switzerland). qRT-PCR was performed using the following thermal conditions: one cycle at 95 °C for 10 min, followed by 2–40 cycles at 95 °C for 15 s, followed by 60 °C for 1 min. The dissociation step was performed at 95 °C for 1 min, 55 °C 30 s, and 95 °C for 30 s. All data were analyzed using the delta ΔCT method. Primer sequences are listed in Appendix A.

### 2.8. RT2 Profiler PCR Arrays

RT2 profiler PCR arrays were performed following the manufacturer’s instructions (RT2 profiler PCR arrays Human Neurogenesis Qiagen Hilden Germany). Briefly, cDNA was diluted with nuclease-free water and added to the RT 2× qPCR SYBR green Master Mix (PAHS-404ZC-24 Qiagen). Then, 25 μL of master mix was added to each well of the human neurogenesis PCR array (PAHS-404ZC-24 Qiagen). Real-time PCR was performed on the one-step QPCR System (Stratagene, La Jolla, CA, USA) following the manufacturer’s thermal profile recommendations: one cycle at 95 °C for 10 min, followed by 2–40 cycles ay 95 °C for 15 s, followed by 60 °C for 1 min (dissociation step at 95 °C for 1 min, 55 °C 30 s, and 95 °C for 30 s). Data were analyzed using DAVID Bioinformatics data analysis web portal. 

### 2.9. Analysis of Real-Time PCR Array Data

Data analysis was performed according to the manufacturer’s instructions [25,31]. Briefly, sample data were normalized to the housekeeping genes ACTB, and the relative expression was subsequently validated using both HPRT and RPL13A. Data analysis was performed by calculating the ΔCt for each gene of interest in the plate (Ct value of gene of interest − Ct value of housekeeping). Ct values >35 were considered negative. The RT2 software averages the triplicate (biological) normalized expression levels for each gene (ΔCt), Before calculating ΔΔCt between control (2D) and spheres (3D) or between control and OCT4 overexpression or control and LDN-19318 treated cells, the RT2 software averaged the triplicate (biological) normalized expression levels for each gene (ΔCt) [32]. The RT2 Profiler PCR Array data analysis software calculates the fold change on the basis of the widely used and agreed upon ΔΔCt method [31]. 

### 2.10. Statistical Analysis of PCR Array Data

The RT2 Profiler PCR Array Data Analysis software performs any statistical analysis of *p*-values using a Student’s *t*-test (two-tailed distribution and equal variances between the two samples) of control group compared to the spheres, OCT4- or LDN-193189 treated group [33].

## 3. Results

### 3.1. Signaling Pathways Regulating Pluripotency of Stem Cells Are Induced in MB Tumorsphere(s)

The nonadherent or ultralow-attachment three-dimensional (3D) culture, also called sphere-forming assay (SFA) is a widely used method to enrich for cancer stem cell-like cells (CSCs) or tumor-initiating cells (TICs) [34]. Hence, to identify signaling pathways connecting TICs with MB, we established 3D cultures according to Gao et al. [29] for three commercially available MB cell lines: DAOY, HDMB-03, and D458 (Figure 1A). We did not observe differences in cell viability between 2D and 3D cultures (Appendix A). Usually, tumorspheres display an enhanced stem-cell marker signature compared to their adherent counterparts. Since CD133 and ALDH1 have been largely recognized as markers of brain TICs [35,36,37], we profiled whether tumorsphere(s) generated from MB cells were enriched for these markers. qPCR analysis showed that CD133 expression was robustly increased under sphere conditions in HD-MB03 and D458 but not in DAOY cells (Figure 1B). In contrast, ALDH1 was enriched only in DAOY but not in HD-MB03 and D458 cells (Figure 1B). We obtained comparable trends also with the classical low-attachment protocol (Appendix A). Despite the stem-cell markers signature being heterogeneous in MB tumorsphere(s), these results suggest that, globally, all cell lines are enriched for cells with TIC capability and stem-cell-like signatures.

To gain better insight into the signaling pathways supporting the maintenance of cells with self-renewal and TICs capability, we performed targeted neurogenesis PCR pathway arrays. This assay is based on transcriptomic analysis of 84 genes manually curated using literature mining. These genes are highly related to the process of neurogenesis, such as neural stem/progenitor maintenance, cell processes, proliferation, differentiation, migration, and maturation into newborn neurons. Growth factors, inflammatory cytokines, and cell adhesion molecules are also represented in this pathway array. Using this approach, we identified the differentially regulated genes in MB tumorsphere(s) (3D) compared to their adherent counterparts (2D) for each cell line (Figure 1C and Appendix A). Gene ontology analysis showed a general heterogeneous gene expression pattern between the three cell lines. Moreover, signaling pathways regulating the pluripotency of stem cells and tumorigenesis were consistently stimulated in all cell lines screened (Figure 1C and Appendix A). To confirm these results, we matched all common differentially regulated genes between the three cell lines, and we found that BMP8B, BMP4, and SOX2 genes, which cluster into signaling pathways regulating the pluripotency of stem cells, were induced in all cell lines (Figure 1D). Overall, these results showed that, beyond a cell line-dependent stem-cell signature, all MB cell lines induced signaling pathways regulating the pluripotency of stem cells when enriched for cells with TIC capability.

### 3.2. The Pluripotent Gene OCT4A Is Induced in MB Tumorsphere(s)

To better understand the genetic pluripotent regulatory network in MB spheres, we carried out qPCR analyses for a panel of pluripotent transcription factors able to reprogram somatic cells back to pluripotency including KLF4, OCT4A, NANOG, SOX2, and c-MYC [17]. Our results showed a significant enrichment in OCT4A, NANOG, and SOX2 in MB tumorsphere(s), which was consistent in all cell lines while OCT4B, KLF4, and c-MYC did not show a consistent trend (Figure 2A and Appendix A). These results indicate a positive correlation between the expression of pluripotent transcription factors OCT4A, NANOG, and SOX2 and TICs in all MB cell lines. 

To investigate the relevance of pluripotent transcription factors in brain cancer, we next analyzed whether they were over-represented across several public datasets generated from patients. Transcriptomic analyses showed that OCT4A, NANOG, and SOX2 were overexpressed in brain tumors (Figure 2B). Later, we restricted our analysis to MB patients. We observed that, across several datasets, SOX2 and NANOG were not significantly overexpressed compared to healthy cerebellum (Figure 2C). Consistently, we did not find a positive correlation between SOX2 and NANOG expression and overall survival in patients (Appendix A). Expression of OCT4A/B and its pseudogenes was significantly increased across multiple patient datasets (Figure 2B). Due to the lack of specific probe, we could not analyze the correlation between OCT4A expression and patient survival within the public databases. However, bioinformatic analysis of published methylation datasets generated from MB patients showed a consistent decrease in DNA methylation at essential regulatory elements of the gene, suggesting that the OCT4A locus in MB is epigenetically poised for transcriptional activation (Appendix A). Consistently, a clinical study performed on a cohort of 50 MB patients showed that high OCT4A levels correlates with a shorter survival and predicts a high risk of relapse [20]. The transcriptional activation of OCT4A observed in MB tumorsphere(s) correlated with increased protein level in all three cell lines (Figure 2D) suggesting that the expression of the pluripotent gene OCT4A is associated with self-renewal and TICs in MB.

### 3.3. Oct4A Enhances Tumor-Initiating Capability and Increases Resistance to Standard Drugs Treatment

To address whether OCT4A promotes TICs in MB, we ectopically overexpressed OCT4A in DAOY, HD-MB03, and D458 cells, and we compared their tumorsphere capability with that of the LACZ control plasmid. The level of OCT4A was analyzed by immunoblots (Figure 3A). The sphere-forming assay showed that forced expression of OCT4A enhanced TICs in all cell lines tested, especially in HD-MB03 (Figure 3B).

To address the impact of OCT4A on the stem-cell marker CD133, we first profiled the expression of CD133 in all cell lines. FACS analysis indicated that CD133 was expressed at the highest level in HD-MB03 cells and was almost undetectable in D458 and DAOY (Appendix A). Consistently, the analysis of CD133 by qPCR showed the same significant trend (Appendix A). To address the impact of OCT4A on CD133 expression, we overexpressed OCT4A in both CD133-positive (HD-MB03) and CD133-negative (DAOY and D458) cell lines. qRT-PCR analyses showed that OCT4A did not modulate CD133 levels in DAOY and D458 cells but efficiently induced the stem-cell markers SOX2 and/or ALDH1A (Figure 3C). In HD-MB03, OCT4A strongly induced CD133 together with c-MYC and SOX2 (Figure 3D). Immunoblots for c-MYC (Appendix A) and FACS analyses for CD133 (Appendix A) showed a similar trend. Consistently, high expression of CD133 and c-MYC correlated with poor survival in patients (Appendix A), suggesting that OCT4A promotes TICs and tumorigenicity potential in group 3 MB cells. Since OCT4A expression mediates the resistance to cisplatin in several cancers [25,26,27], we first analyzed the cytotoxic effect of cisplatin in the group3 HD-MB03 MB cell lines. Parental cell lines were treated with 5 µM cisplatin for 48 h to compare the response of parental and OCT4-expressing cells. MTT assays suggested that OCT4A is associated with reduced cytotoxicity to cisplatin (Appendix A). Later, we extended our analysis to chemotherapies more commonly used in clinical setting such as etoposide and vincristine. Parental cells were treated with 10 nM–100 μM etoposide to find the related IC50 dose (Appendix A) and used subsequently to compare the response of parental and OCT4A-expressing cell lines. MTT assays showed that OCT4A is associated with reduced cytotoxicity to etoposide in all cell lines tested (Figure 3D). Consistently, OCT4 overexpression reduced vincristine sensitivity (Appendix A), supporting a link between OCT4 expression and chemotherapy sensitivity.

### 3.4. OCT4A Induces BMP4-ALK2/3 Signaling Pathway in MBs

To further elucidate which signaling pathways are induced downstream to OCT4A that could be connected with TICs, we used targeted neurogenesis pathway arrays (see Section 2). We overexpressed OCT4A in all three adherent parental cell lines and we compared them to parental adherent counterpart. Gene ontology analysis showed overall a heterogeneous gene expression pattern between cell lines (Figure 4A). However, similarly to parental tumorsphere(s), signaling pathways regulating pluripotency of stem cells and tumorigenesis were consistently induced in all three OCT4A-expressing cell lines (Figure 4B and Appendix A) suggesting that OCT4A could promote pluripotency of stem cells and tumorigenesis in MB through induction of the BMP4 signaling pathway.

Next, we matched the differentially regulated genes in parental spheroids (2D/3D) with those differentially regulated upon OCT4A overexpression (WT/OCT4) for each cell line. Gene ontology analysis showed that OCT4A-expressing HD-MB03 cells had the highest match with the tumorsphere counterpart (Appendix A). Using this criterion, we found only seven overlapping genes across all cell lines, namely, BMP4, ADORA2, APOE, CXCL1, GDNF, HEY1, and IL3 (Appendix A). Among these genes, BMP4 was the only gene clustering into pathways regulating pluripotency of the stem cells (Figure 4B). To evaluate whether BMP4 and its receptors ALK2/3 promote stem-cell traits in MB, we first verified whether the signaling is induced following a dose–response evaluation for BMP4. Immunoblots for pSmad1/5, a downstream readout of ALK2/3 signaling, showed that the BMP4–ALK2/3 pathway is functional in HD-MB03 cells (Figure 5A). HD-MB03 cells cultured with conditioned medium supplemented with 100 nM BMP4 had an increased number of spheres (Figure 5B) suggesting that BMP4–ALK2/3 signaling sustains TICs in MB cell lines. Consistently, FACS analysis showed that BMP4 but not TGF-β increased the size of the CD133+ cell population (Figure 5C and Appendix A), arguing that BMP4–ALK2/3 signaling might support stemness in MB.

### 3.5. Inhibition of ALK2/ALK3 Signaling Blocked Migration, Reduced TICs, and Increased Cytotoxicity to Chemotherapy in MB

To evaluate the relevance of targeting ALK2/3 receptors in MB, we performed RNA expression analysis of published datasets generated from patients. We found that ALK2/3 are overexpressed in MB patients compared to a healthy cerebellum, supporting the relevance of targeting ALK2/3 signaling in MB (Appendix A). Next, to interfere with ALK2/3 activity, we used the LDN-193189 inhibitor, since it displays beneficial effects in preclinical models of diffuse intrinsic pontine glioma [38].

Immunoblot analyses for the downstream effector p-Smad1/5 indicated that LDN-193189 efficiently blocked the signaling at the basal level and following BMP4-mediated induction (Figure 5D). To verify whether the inhibition of ALK2/3 signaling had any effect on stem-cell markers, we treated WT and OCT4-expressing HD-MB03 cells with LDN-193189. qPCR analyses showed that LDN-193189 reduced the level of CD133 and c-MYC expression in WT and OCT4-expressing cells, while SOX2 and ALDHA1 remained unchanged (Figure 5E and Appendix A). FACS analyses confirmed that LDN-193189 decreased the size of the CD133+ population (Figure 5F), suggesting that the ALK2/3 inhibitor could decrease the cells with TICs in MBs. To verify this hypothesis, we performed sphere assays in conditioned media supplemented with LDN-193189, and we found that LDN-193189 was able to revert both OCT4-dependent and -independent stem-cell traits in MB (Figure 5G and Appendix A). Next, we addressed whether BMP4–ALK2/3 signals through the canonical p-Smad pathway or via the non-canonical pathway. For this aim, we checked whether OCT4A over-expression could modulate the phosphorylation of pSmad1/5. Western blot analysis did not show induction of pSmad1/5 (Figure 6A). However, OCT4A overexpression decreased the level of p-ERK in an LDN-193189-dependent manner (Figure 6B), suggesting that BMP4 signals through the non-canonical pathway in HD-MB03.

Since, in normal stem cells, inhibition of p-ERK promotes pluripotency and constrains cell differentiation, we tested whether LDN-193189 could promote cell differentiation in MB. To address this question, we performed targeted neurogenesis pathway arrays to identify the signaling pathways differentially regulated upon LDN-193189 related to untreated cells. Gene ontology analyses indicated that inhibition of ALK2/3 signaling pathway induced pathways involved in neurogenesis and cell differentiation, and it concomitantly down-regulated pathways supporting pluripotency of stem cells and pathways in cancers (Figure 6C,D and Appendix A). Together, these results suggest that LDN-193189 could reduce the pool of TICs by promoting cell differentiation. It is known that BMP4–ALK2/3 signaling promotes cell migration of cancer cells, which is an important feature of chemotherapy resistance [39]. Therefore, we checked whether LDN-193189 might have a beneficial effect on MB cell migration. Boyden chamber assays showed a strong reduction in cell migration in DAOY and HD-MB03 (Appendix A). Furthermore, LDN-193189 increased the cytotoxicity to chemotherapies (Figure 6E). Overall, this study highlights the role of pluripotency-related BMP4 signaling in supporting the maintenance of the CSC pool in MB. The BMP4–ALK2/ALK3 signaling pathway promotes the pluripotency genetic network at least through inhibition of p-ERK. In turn, the induction of the pluripotency genetic network also induces BMP4 signaling and generates a regulatory feedback loop that maintains the CSC pool within the tumor.

## 4. Discussion

MB is an embryonic tumor of the cerebellum with a pronounced stem- or progenitor-like signature [1]. CSCs possess higher tumorigenic potential as compared to differentiated cells. They play an important role in promoting tumorigenesis through their crucial role in cancer initiation, progression, metastasis, and relapse. CSCs are also more resistant to chemotherapies, are capable of clonal long-term repopulation, and drive tumor relapse. Quiescence is one of the self-protective mechanisms of CSCs allowing long-term survival through adaptive responses to environmental stress. The quiescent state of CSCs protects these cells from anti-proliferating agents such as traditional radiotherapy and chemotherapy. However, signs of differentiation into neutrons, glia, and other cell types have been detected in MB [40], suggesting that these cells potentially might retain their ability to differentiate. Cell differentiation inversely correlated with tumorigenicity and drug resistance. Hence, cell differentiation could be used as a therapeutic approach for MB cure. To date, it is unclear whether cancer cells isolated using cell surface markers or side populations represent TICs. Indeed, universal strategies to identify and isolates CSCs using specific surface markers have not been established due to the inter-tumor heterogeneity. Consistently, MB cells showed a heterogeneous stem-cell marker profile despite all lines being able to form TICs. Therefore, functional assays such as sphere formation assays (SFA) rather than marker selection are the most reliable in vitro tool to enrich for cancer stem-like cells (CSCs) or tumor-initiating cells (TICs) [22,23,41,42]. In this study, by combing tumorsphere(s) forming assays with targeted neurogenesis PCR pathway arrays, we found a consistent induction of signaling pathways regulating pluripotency of stem cells and tumorigenesis in all MB cells screened. The induction of pathways regulating pluripotency of stem cells suggested that the expression of embryonic pluripotent transcription factors could promote enrichment of TICs in MB. Therefore, we decided to profile the expression of a panel of pluripotent genes in MB tumorsphere(s). Among these, OCT4A, SOX2, and NANOG were consistently enriched in tumorsphere(s) generated from all cell lines. While SOX2 and NANOG expression did not correlate with patient survival, high OCT4A was associated with shorter overall survival in MB patients and predicted high risk for relapse [25]. OCT4A is a transcription factor that plays an indispensable role in self-renewal and pluripotency of embryonic stem cells [18] and CSCs [19,20,21]. High OCT4A expression correlated with poor clinical outcome in several cancers including medulloblastoma [22,23,25,26,27,43,44,45], and it has been shown that OCT4A can drive relapse and repopulate tumors after release of chemotherapy [46,47]. Da Silva et al. previously reported that forced expression of OCT4 drove tumorigenecity by enhancing tumorsphere generation in MB cell lines [22]. Accordingly, OCT4 increased TIC capability and reduced cytotoxicity of chemotherapies. Preclinical/clinical studies correlated OCT4 with poor response to chemotherapy in several cancers including medulloblastoma [25], further validating our cell model. These observations suggested that OCT4 expression can serve as a prognostic factor of tumor aggressiveness and a predictive marker of resistance to treatments. This characteristic is instrumental to anticipate resistance and to administer alternative therapies. However, the mechanism through which OCT4A promotes tumorigenesis has not been yet elucidated. To gain insights into the mechanisms that support CSCs in MB, we dissected the signaling pathways induced downstream to OCT4 activation. We identified the key role of the BMP4–ALK2/3 signaling in supporting CSCs. Consistently, ALK2/3 signaling was enriched in all MB tumorsphere(s) independently of the specific stem-cell marker profile. Activin receptor-like kinase-2 (ALK2) is a type I bone morphogenic protein (BMP4) receptor which controls the development of bone, heart, brain, and reproductive systems. The BMP4 signaling cascade, together with WNT, FGF, and Notch pathways, constitutes the stem-cell signaling network in pluripotent stem cells [48]. BMP4–ALK2 signaling has been associated with adult stem cells [49]. BMPs are deregulated in several types of cancer; however, their role remains controversial [50]. For instance, in breast cancer, BMP4 promotes tumorigenesis [51], while another study showed, paradoxically, a tumor-suppressing function of BMP4 [52]. Similar to normal stem cells, BMP4 has been associated with CSCs in several tumors including breast cancers [53] and glioblastoma [54]. However, its role in MB cancer stem cells has never been investigated. In this study, we showed that BMP4–ALK2/3 signaling enhances TICs and stem-cell traits in MB cells. 

In turn, inhibition of this signaling decreased TICs capability and correlated with reduced expression of stem-cell markers. The BMP4–ALK2/3 signaling pathway works through inhibition of the non-canonical p-ERK pathway rather than through induction of the canonical p-Smad1/5 pathway. We reasoned that, similarly to normal stem cells, inhibition of ERK activity could promote pluripotency and constrain differentiation in MB. Consistently, inhibition of ALK2/3 signaling using the LDN-193189 inhibitor increased ERK activity, reduced stem-cell traits, and promoted cell differentiation. Furthermore, LDN-193189 increased the cytotoxic effects of chemotherapies and decreased the migration potential of MB cells. The effectiveness of LDN-193189 was recently shown in preclinical mouse models of glioma [38]. However, a mechanistic understanding on how LDN-193189 is beneficial for brain cancer treatment is not very well delineated. In this study, we found that LDN-193189 reduced stem-cell traits and promoted cell differentiation, suggesting that inducing cell differentiation could be beneficial for MB treatment. A recent paper showed that the BMP4 signaling pathway promotes trans-differentiation of tumor cells in relapsed medulloblastoma, supporting the argument that this pathway could acts in maintaining the CSC pool in both primary and relapsed tumors [55].

## 5. Conclusions

Differentiation therapy, in cancer, aims to force the more undifferentiated cells to resume the process of maturation as a strategy for drug intervention. The prototype example of agents promoting differentiation as a useful therapy is acute promyelocytic leukemia with retinoic acid. Although differentiation therapy does not destroy the cancer cells, it restrains their proliferation and improves the efficacy of conventional therapies. Considering that 70% of patients are cured following conventional treatments, the major concern is related to the choice of introducing this approach in the first line or following relapses. Differentiation therapies could be administered in the first line and combined with standard-of-care chemotherapies to reduce highly detrimental treatment-dependent side-effects. Since relapses are 100% fatal, differentiation therapies may improve the efficacy of second-line treatments by reducing the CSCs pool. Despite much more research being needed to understand the mechanisms underlying CSC differentiation in brain cancers, this innovative strategy must be further evaluated as a promising therapeutic strategy to be implemented in the clinic to prolong remissions of MB.

## Figures and Tables

**Figure 1 cancers-14-02095-f001:**
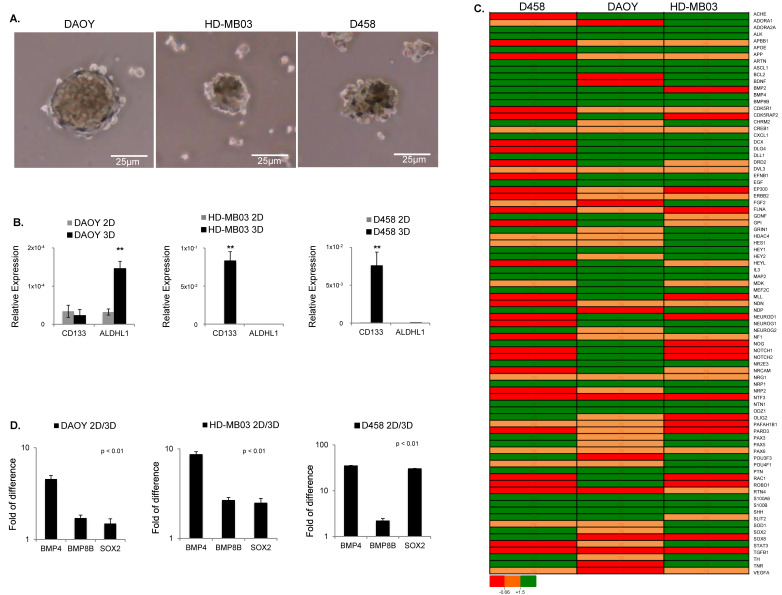
MB tumorsphere(s) generated from cell lines as indicated. (**A**) Representative images of spheroids generated from DAOY, HD-MB03, and D458 cell lines as indicated. (**B**) qRT-PCR analysis of brain CSC-related markers showing the relative expression of CD133 and ALDH1 in adherent (2D) and spheroid (3D) cells generated from DAOY, HD-MB03, and D458 relative to GAPDH (n = 3 biological replicates). ** *p* < 0.01, according to two-tailed paired Student’s t-test. Data are the mean ± SD. (**C**) Heatmap of RT2 profiler PCR arrays showing the differentially regulated genes in all three cell lines as shown. The gene expression level is shown as the fold change in 3D relative to 2D and normalized to the expression of housekeeping genes ACTB. ** *p* < 0.01. (**D**) RT2 profiler PCR array analysis for DAOY, HD-MB03, and D458 showing the gene expression level of BMP4, BMP8B, and SOX2 expressed as the fold change in tumor sphere(s) (3D) compared to (2D) and normalized with the housekeeping genes ACTB. *p* < 0.01 Data are the mean ± SD.

**Figure 2 cancers-14-02095-f002:**
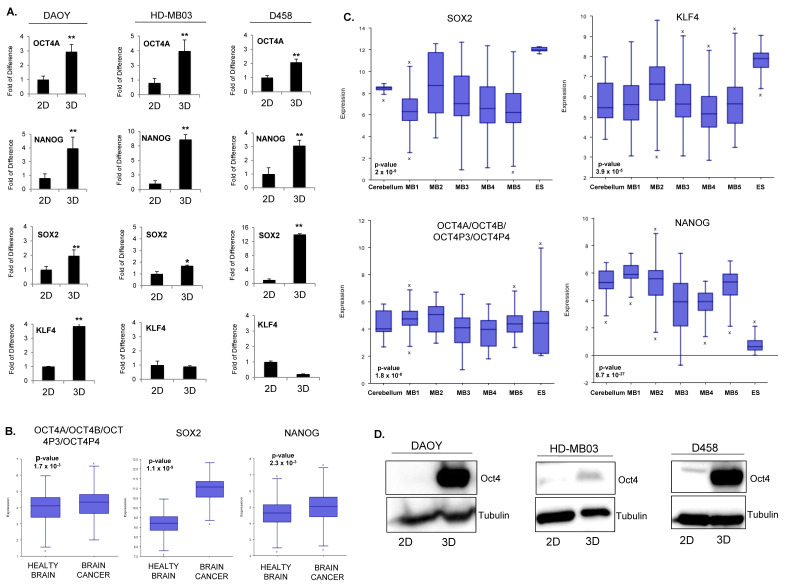
Pluripotent regulatory network in MB tumorsphere(s). (**A**) qPCR analysis of pluripotent transcription factors OCT4A, NANOG, SOX2, and KLF4 in spheres (3D) generated from DAOY, HD-MB03, and D458 lines relative to adherent counterpart (2D). * *p* < 0.05; ** *p* < 0.01; according to two-tailed paired Student’s t-test. Data are the mean ± SD. (**B**) Box–dot plot showing the expression level of OCT4A/OCT4B/OCT4P3/OCT4P4, SOX2, and NANOG in healthy brain and brain cancer. *p* < 0.001, according to one way analysis of variance (ANOVA). Data are the mean ± SD. (**C**) Box–dot plot showing the expression level of SOX2, KLF4, OCT4A/OCT4B/OCT4P3/OCT4P4, and NANOG in healthy cerebellum (cerebellum n = 9 from Roth database) and MB patients (M1: (n = 76) Gilbertson database, M2: (n = 51) from de Boer database, M3: (n = 223) from Pfister database, M4: (n = 57) from Delattre database, M5: (n = 62) from Kool database and embryonic stem cells, ES: Neuronal embryonic stem cells (n = 6) from Viale database). *p* < 0.001, according to One way analysis of variance (ANOVA). Data are the mean ± SD. (**D**) Western blot analysis showing the protein level of OCT4 in adherent (2D) and spheroid (3D) cells for each cell line as indicated. Tubulin expression level was used as a loading control. The original blots could be found in the Appendix A.

**Figure 3 cancers-14-02095-f003:**
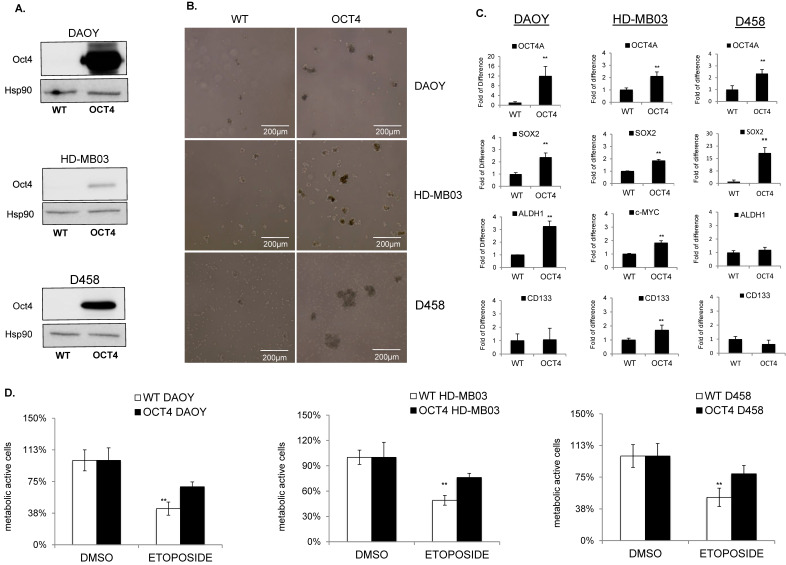
OCT4A enhances TICs and resistance to chemotherapy. (**A**) Immunoblot analysis shows the expression level of OCT4 protein after overexpression in DAOY, HD-MB03, and D458 cells. Hsp90 was used as a loading control. (**B**) Representative images of tumorsphere(s) generated from wildtype (WT) (left panel) and OCT4-expressing (OCT4) cells (right panel) generated from DAOY, HD-MB03, and D458 cell lines as indicated. (**C**) qPCR analyses showing the expression level of OCT4A, CD133, SOX2, and ALDH1 in wildtype (WT) and OCT4-overexpressing (OCT4) cell lines as indicated. ** *p* < 0.01, according to two-tailed paired Student’s t-test. Data are the mean ± SD. (**D**) Cytotoxic effect analyzed in MTT assays of etoposide in parental (WT) and OCT4-overexpressing (OCT4) HD-MB03, DAOY, and D458 cells related to DMSO treated counterpart (DMSO). The original blots could be found in the Appendix A.

**Figure 4 cancers-14-02095-f004:**
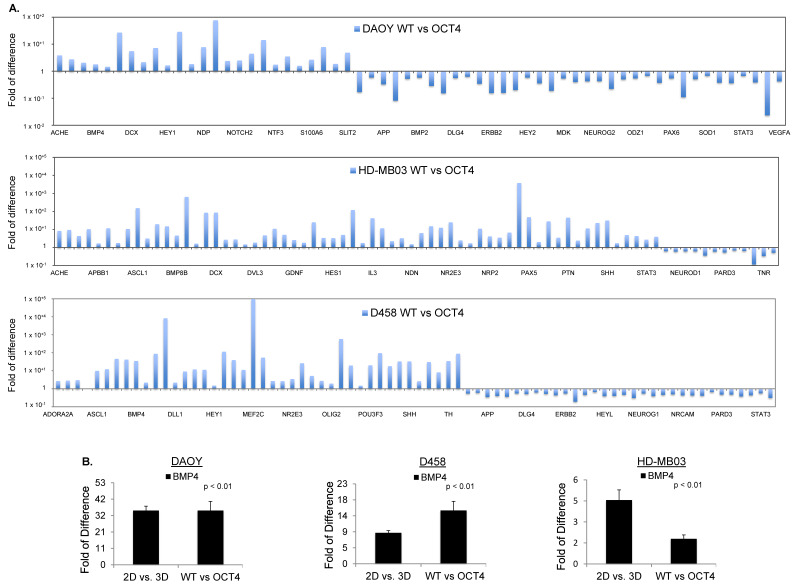
OCT4A over-expression induces BMP4 signaling in MB cells. (**A**) RT2 profiler PCR arrays performed in DAOY, HD-MB03, and D458 cells showing the expression level of differentially regulated genes in OCT4-overexpressing (OCT4) cells related to parental counterpart (WT) cells and normalized to the expression of housekeeping gene ACTB. (**B**) RT2 profiler PCR arrays of BMP4 level performed in OCT4-overexpressing DAOY, HD-MB03, and D458 cells related to parental counterpart (WT) cells and normalized to the expression of housekeeping gene ACTB.

**Figure 5 cancers-14-02095-f005:**
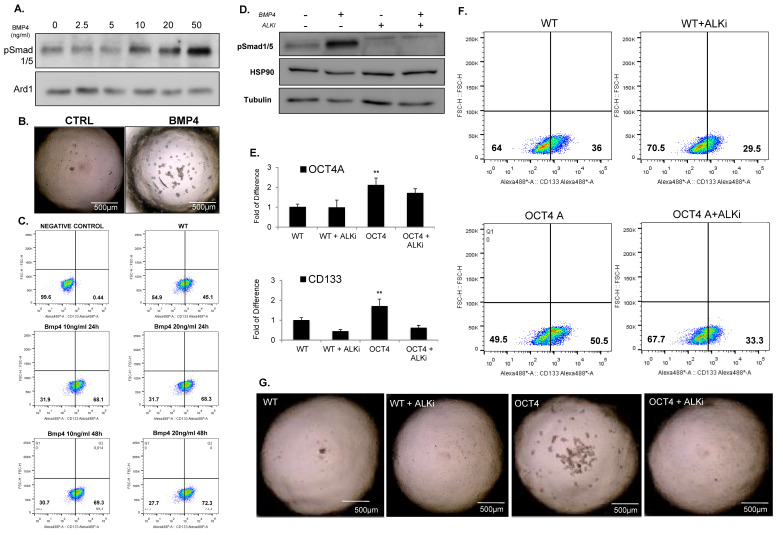
BMP4 signaling promotes TICs and stem cell traits. (**A**) Immunoblot analysis showing the protein level of phosphorylated Smad1/5 in HDMB03 cells exposed to increased dose of BMP4 from 0 to 50 ng/mL. Tubulin was used as a loading control. (**B**) Representative images of tumorsphere(s) generated from HD-MB03 untreated (CTRL) (left panel) and treated with 50 ng/mL of BMP4 (BMP4) (right panel). (**C**) FACS analysis showing the expression of CD133 in HD-MB03 exposed to 10 ng/mL or 20 ng/mL of BMP4 for 24 h or 48 h. (**D**) Immunoblot showing the protein level of phosphorylated Smad1/5 in HD-MB03 cells treated or untreated with 20 ng/mL of BMP4 and ALK2/3 inhibitor (LDN 198836) for 24 h. (**E**) qPCR analysis showing the expression of stem-cell markers OCT4A, and CD133 in parental HDMB03 (WT) and OCT4-expressing HD-MB03 (OCT4) cells treated or untreated with ALK2/3 inhibitor (LDN 198836) for 24 h. ** *p* < 0.01, according to two-tailed paired Student’s t-test. Data are the mean ± SD. (**F**) FACS analysis for CD133 in parental HD-MB03 (WT) and OCT4-expressing HD-MB03 (OCT4) cells treated or untreated with ALK2/3 inhibitor (LDN 198836) for 24 h. (**G**) Representative images of tumorsphere(s) generated from parental (WT) and OCT4-expressing (OCT4) HD-MB03 cells cultured with or without ALK2/3 inhibitor (LDN 198836). The original blots could be found in the Appendix A.

**Figure 6 cancers-14-02095-f006:**
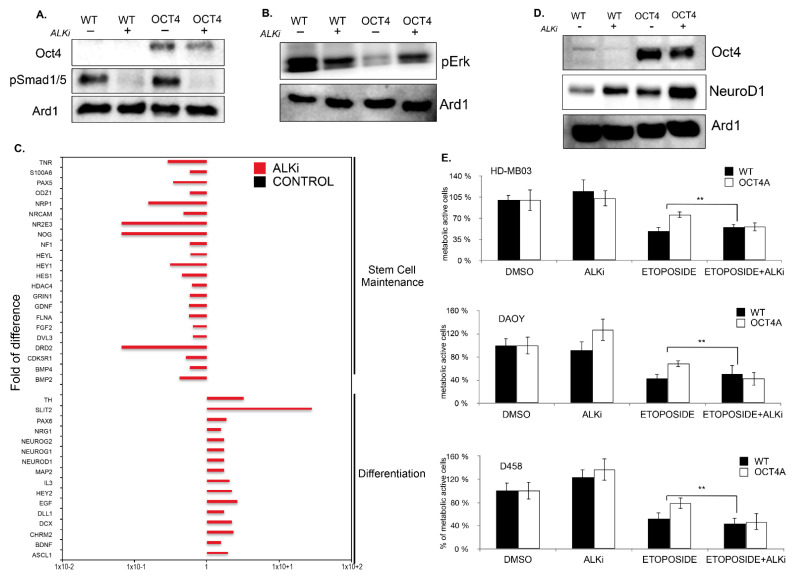
The ALK2/3 inhibitor, LDN 198836, promotes MB cell differentiation and cytotoxicity to chemotherapy. (**A**) Immunoblot analysis showing the level pSmad1/5 in parental HD-MB03 (WT) and OCT4-overexpressing (OCT4) cells treated or untreated with ALK2/3 inhibitor (LDN 198836). (**B**) Immunoblot analysis showing the p-ERK levels in parental HD-MB03 (WT) and OCT4-overexpressing (OCT4) cells treated or untreated with ALK2/3 inhibitor LDN 198836. (**C**) RT2 profiler PCR arrays performed in HD-MB03 cells showing the differentially regulated genes upon treatment with ALK2/3 inhibitor (LDN 198836) compared to untreated parental (WT) cells and normalized to the expression of housekeeping gene ACTB. ** *p* < 0.01. (**D**) Immunoblot analysis showing the level of NeuroD1 in parental HD-MB03 (WT) and OCT4-overexpressing (OCT4) cell line treated or untreated with ALK2/3 inhibitor (LDN 198836). (**E**) Cytotoxic effect measured in MTT assay of parental (WT) and OCT4-overexpressing (OCT4) HD-MB03, DAOY, and D458 cells treated with etoposide and with etoposide plus ALK2/3 inhibitor (LDN 198836) related to DMSO-treated counterpart (DMSO) cells. The original blots could be found in the Appendix A.

## Data Availability

For patients’ expression data (Figure 2B,C and Appendix A) and DNA methylation analysis, we used public datasets deposited on the “R2: Genomic Analysis and Visualization Platform”: https://hgserver1.amc.nl/cgi-bin/r2/main.cgi (accessed on 12 February 2022); the provided tools were used to generate all graphics and statistical analysis. Kaplan–Meier curves, for overall survival analysis of patients, were generated using the R2 Cavalli database (https://hgserver1.amc.nl/cgi-bin/r2/main.cgi (accessed on 12 February 2022)). For KEGG pathway analysis, we used the portal https://david.ncifcrf.gov/ (accessed on 12 February 2022).

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
