# Peer review of "Inhibiting ALK2/ALK3 Signaling to Differentiate and Chemo-Sensitize Medulloblastoma"

_cancers, 2022, doi:10.3390/cancers14092095_

Round 1

Reviewer 1 Report

The hypothesis of tumor-initiating cells (TIC) or cancer stem cells (CSCs) is not new. It was shown before that after therapies some “survived” cancer cells can cause the tumor relapse with pronounced drag resistance and even worst outcome than the original. However, as truly was stated by authors, there is a big challenge to identify signaling pathways that support CSCs maintenance, effectively target them and CSCs as well. In the present study, the authors use appropriate methods and approaches showing that BMP4-ALK2/3 signaling enhances TICs and stem cell traits in medulloblastoma cells. Another important finding was that inhibition of ALK2/3 signaling using the LDN-193189 inhibitor increases ERK activity reduces stem cell traits and promotes cell differentiation. Moreover, LDN-193189 increases the cytotoxic effects of chemotherapies. The proposed strategy of targeting CSCs by induced differentiation is very promising as a therapeutic strategy for cancer treatment.

Overall, very important and interesting study and deserves to be published in “Cancers” with minor revisions.

Downside: Animal experiments are missing.

Some suggestions:

Please, increase the font size in following Figures:  Fig.1C; Fig.4A; Fig.6C; Fig.S1(DEF); Fig.S3B

Some typos have been noticed:

R.28 – ”… all the screened MB cells screened.”     should be “… all the screened MB cells.”

R.52 – “… including brain, [8, 10, 11]. Cur-“  should be “… including the brain [8, 10, 11]. Cur-“

R.87 and R.140 – “CO2”  should be    “CO2

R.92 – “… >75μM of diameter”   should be   “… >75μm of diameter”

R.119 –“… 1H”    should be    “… 1h”

R.251 – “… has a loading control.”    should be  “… as a loading control.”

R.26; 443; 449; 450 – “… tumor-sphere(s)” everywhere else “… tumorsphere(s)” [make it the same on your preference]

Thank you!

Reviewer 2 Report

In this paper, entitled “Inhibiting ALK2/ALK3 Signaling to Differentiate and Chemo-sensitize Medulloblastoma” by Doria Filipponi and Gilles Pagès, it is suggested that inhibition of BMP4-ALK2/ALK3 signaling results in differentiation of medulloblastoma (MB) stem cells and that MB cells become more sensitive to chemotherapy. The authors also suggest that this method could provide a new treatment modality. The study is well written, and easy to follow. The concept of targeting tumor stem cells with therapies aiming at differentiating cells has shown promising results in experimental models. Still, clinical data is missing. In order to bring new knowledge to the field, suggested treatment modality should also be tested in MB organoids. Or at least in spheroids originating from MB cells not previously cultured adherently in FBS, as these models better mimic the clinical situation.         

MAJOR

GENERAL CONCEPT

Do you really enrich for cancer stem cells in a cell line that has been long-term cultured in serum containing medium, only by changing conditions such as removing adherence/FBS and adding growth factors? Or, is it a general phenomenon for the bulk of tumor cells to form spheroids, and change phenotype when transferred from FBS-containing medium to serum free stem cell medium?

Suggestion: To bring new originality and novelty to the field authors should test their concept in primary MB organoids, that better mimic tumor heterogeneity than cell line-derived spheroids. Or at least in MB cells cultured in stem cell medium from establishment, to avoid in vitro artefacts caused by long-term culturing in FBS and adherent growth.

Fig. 1/Fig.3/Fig. S6

Pictures of spheroids - Spheroids looks more like loose cell clusters formed by lack of adherence rather than compact spheres derived from tumor stem cells. In addition, the center looks very dark, as an indication of cell death due to lack of nutrition.

Suggestion: Is it possible to seed out single MB cells in wells for testing their sphere forming capacity, and then use these spheroids in the various assays?

Fig. 1 Increase of BMP4.

Is the increase of BMP4 in spheroids (3D) compared to adherent cells (2D) related to “stem-ness” or to the fact that cells in spheroids become more senescent due to lack of nutrition?

Suggestion: Compare 2D/3D cell viability. The role of BMP4 expression in tumors is dual, and BMP4-treatment has been utilized as a way to differentiate tumor stem cells. The paper would benefit of including a few references and a discussion about BMP4 and its opposing actions.

MINOR

Fig3 E/Fig5 C

It is difficult to see difference in CD133 expression between OCT4 expressing and WT cells, or between BMP treated and control cells.

Suggestion: Arrange data in the same FACS-plot, or present “area under the curve”. Why was not an Isotype ctrl included?

Fig 6E

In Fig 6E it is shown that only in the OCT4 expressing cells the ALK inhibitor sensitize the cells for chemotherapy. However, the cell viability effect by Etoposide+ALK inhibitor looks the same in WT and in OCT4 expressing cells, if etoposide alone is excluded. From this figure, it is not clear if it matters whether the cells are WT or OCT4 expressing “stem cells”, as long as the end result is a reduction of cell metabolic activity. Could this be explained further?

Reviewer 3 Report

This is overall a very interesting and well designed study that will controbute to our understanding of the relationship between OCT4 and medulloblastoma properties. I like the set of experiments the authors decided to perform. Priritizing the use of cell lines (over primary cultures or in vivo models) does not represent  a limitation within the context of this study.

If I can point an aspect that could be improved, I am not sure the relevant current literature on OCT4 and medulloblastoma, and on the relationships among stemness markers, differentiation, and medulloblastoma properties and recurrence, has been fully covered in the Introduction and Discussion sections.    

Reviewer 4 Report

the manuscript present multiple issues: many figures must be fixed, experiments must be extended on all cell lines and maybe same in vivo experiments would be necessary and results presented ina more clear way.

The major concern is the fact that there is a paper already publish in which has been demonstrated that  BMP stimulated transdifferentiation of MB cells and its signaling Inhibition suppresses the transdifferentiation of medulloblastoma cells. They demonstrate in vivo.

https://doi.org/10.1084/jem.20202350

In my opinion the presented work lack of novelty and is not sufficient to be take in consideration for Cancers publication

Author Response

the manuscript present multiple issues: many figures must be fixed, experiments must be extended on all cell lines and maybe same in vivo experiments would be necessary and results presented in a more clear way.

--Reply: We thank the reviewer for critical feedback on the manuscript. We now fixed the figures as much as possible and we extended key experiments on all cell lines. All experiments presented in Figures 1-4 and part of Figure 6 have been performed on all cell lines. In Figure 3 we added in the revised version results on D458 cells as well and sensitivity to etoposide in all cell lines. These results confirmed previously clinical results linking OCT4 overexpression with poor prognosis (Rodini et al., 2020) and reduced sensitivity to chemotherapy (Mohiuddin et al., 2020). In Figure 6E we included the cytotoxic effect of ALKi and we extend these results to all cell lines. According to that, we also fixed the presentation of results.

The major concern is the fact that there is a paper already publish in which has been demonstrated that BMP stimulated transdifferentiation of MB cells and its signaling Inhibition suppresses the transdifferentiation of medulloblastoma cells. They demonstrate in vivo.

--Reply: We thank the reviewer to point out this paper that strongly reinforce the significance of our findings. In the paper published on JEM in September 2021, Gua et al., highlight the role of BMPB4 in promoting trans-differentiation in relapsed tumors in a SHH-driven mouse genetic model.

To bring new knowledge that could potentially improve the standard of care of medulloblastoma, the focus of our work was to identify, with the aim to target, signaling pathway that could play a role in maintenance of CSCs (rather than target CSCs drought markers selections). In turn, inhibition of BMP4 signaling promoted cell differentiation and improve the efficacy of chemotherapy. It is known that chemotherapy might promotes de-differentiation or reprogramming of cancer cells to CSCs generating a loop that can continuously support the CSCs pool during the onset of the treatment. However, while in our study we did not examine the effect of the inhibitor on relapsed tumor, the author showed that in mice, inhibition of BMP4 signaling constrain such phenomena supporting the role of BMP4 in the maintenance of both intrinsic and dynamic CSCs pool. However, this observation has been made using a specific mouse genetic model (SHH-driven) that do not represent the most aggressive types of human MB and do not take in account the tumor heterogeneity between different subgroups. In our study, we decided to focus on human cell models representing different subgroup of MD and matching into the group of high-risk cases. Our findings showed that BMP signaling is involved in maintenance of the stem cell phenotype in primary tumor. Therefore, our results combined with those of Guo et al strongly suggest that inhibition of BMP signaling could be used at the diagnosis or following relapse on standard of care. Hence, testing at the diagnosis the activation of the BMP pathway would serve to select eligible patients to the treatment.

1) It is also important to point-out that the paper of Guo et al was performed on a mouse genetic model of MB. The tumorigenesis mechanism observed in mice, despite very useful, not necessary always mimic the one of human. Our study on human cells showed the relevance of ALKi in targeting the human CSCs pool while Guo et al showed the relevance in reducing trans-differentiation during chemotherapy in mice. Together, the results of Guo et al and our study reinforce the notion that targeting BMP signaling could be a relevant approach for the treatment of MB.

2) We also believe in the relevance of repositioning the ALK inhibitor used in a clinical trial for the treatment of MB patients.

In my opinion the presented work lack of novelty and is not sufficient to be take in consideration for Cancers publication.

--Reply: Our experiments and those of Guo et al show the relevance of repositioning the ALK inhibitor for the treatment of MB patients. It will be encouraging as next preclinical follow-up to test BMP4 inhibitor in patients-derived model of medulloblastoma since these 2 complementary studies show the robustness of targeting the BMP signaling both at the diagnosis (our study) and following relapse (Paper of Guo et al and our study). Moreover, by combining the two drugs (ALKi and etoposide) at lower concentrations may limit detrimental side effects but may maintain therapy efficacy and allow a long-term control of the disease.

Round 2

Reviewer 4 Report

 My opinion is not changed